# Adverse childhood experiences, stress impact, and well-being in deaf and hard of hearing adolescents and adolescents with developmental language disorders in special secondary education

L. C. Martijn[1,2]*, D. Hermans[2,3], H. E. T. Knoors[2,3], C. T. W. M. Vissers[2,3]

**1** Stichting VierTaal, VierTaal College Amsterdam, Amsterdam, Noord-Holland, the Netherlands, **2** Behavioural Science Institute, Radboud University, Nijmegen, Gelderland, the Netherlands, **3** Koninklijke Kentalis, Utecht, Utrecht, the Netherlands

* Len.martijn@ru.nl

## Abstract

Many deaf and hard of hearing (DHH) adolescents and adolescents with developmental language disorders (DLD) face communication problems (CP) that may increase their vulnerability to adverse childhood experiences (ACEs). ACEs have been shown to affect physical and mental health, and in particular, the accumulation of ACEs is associated with high stress levels, which can have detrimental effects on well-being. This study examined the prevalence of ACEs and their impact on stress and well-being in adolescents with CP (DHH, DLD) compared to a reference group.127 adolescents with CP (32 DHH adolescents and 95 adolescents with DLD) in special secondary education were compared to a reference group of 86 adolescents in mainstream secondary education. T-tests, chi-square tests, and proportion tests were used to compare all groups on ACEs, stress impact, and well-being. A mediation analysis tested the indirect effect of stress on the relationship between ACEs and well-being. Adolescents with CP reported significantly more (accumulations of) ACEs *(p < .05, p.001)*, higher stress impact *(p < .001)*, and lower well-being *(p < .05)* than the reference group. The origin of communication problems (DHH versus DLD) appeared to induce no differences between groups. Mediation analysis revealed that ACEs indirectly influenced adolescents' well-being through their effect on stress impact *(p < .001)*. Adolescents with CP in special secondary education are highly vulnerable to encountering ACEs, which increases the risk of experiencing high stress levels that detract from their well-being. Therefore, structural screening of this group for stress impact and well-being should be mandatory.

**Data availability statement:** Supplementary materials, original anonymized data files, and outputs are available for open access in a repository named: 'Adverse childhood experiences, stress impact, and well-being in deaf and hard of hearing adolescents and adolescents with developmental language disorders in special secondary education' at https://doi.org/10.34973/dgjh-ja34 as assigned by Radboud University Nijmegen.

**Funding:** All research-related expenses (travel expenses for the first author, 10 euros participant rewards, hiring independent sign language interpreters) were covered by Stichting VierTaal in Amsterdam, the Netherlands. Authors' salaries were provided by their employing institutions: H.E.T. Knoors, C.T.W.M. Vissers, and D. Hermans from Koninklijke Kentalis in Utrecht, the Netherlands; L. Martijn from Stichting VierTaal in Amsterdam, the Netherlands. Both funders (Koninklijke Kentalis in Utrecht and Stichting VierTaal in Amsterdam) had no role in the study design, data collection and analysis, decision to publish, or manuscript preparation. Funding order: 1. Stichting VierTaal, Amsterdam, the Netherlands 2. Koninklijke Kentalis, Utrecht, the Netherlands.

**Competing interests:** The authors have declared that no competing interests exist.

## Introduction

Adverse childhood experiences (ACEs), such as child maltreatment, are associated with long-term negative outcomes for mental health and overall well-being, particularly when multiple ACEs accumulate [1–3]. Children and adolescents with disabilities appear to have an increased risk of experiencing ACEs compared to their non-disabled peers [4–6]. There is evidence that children with communication problems (CP), such as those caused by hearing loss or developmental language disorders (DLD), are also at higher risk of encountering ACEs [7–13]. Despite these findings, there seems to be a lack of studies examining a wide range of ACEs, the accumulation of these ACEs, and the impact of this accumulation on stress levels and overall well-being in DHH populations and those with DLD. This is precisely the focus of our study.

To contextualize this focus, it is essential to understand what ACEs entail and why they matter. ACEs encompass a range of potentially traumatic events occurring during childhood and adolescence, such as maltreatment, household dysfunction, and economic hardship. ACE research has demonstrated that childhood adversities significantly contribute to negative physical and mental health outcomes throughout life [1–3]. While exposure to a single adverse event can elicit significant stress, there is a well-established theoretical framework suggesting that the cumulative burden of multiple ACEs is most strongly associated with long-term detrimental effects on health and well-being [3,14]. This association between ACE accumulation and adverse health and well-being outcomes has been well documented in research [1,15]. Some researchers contend that, in addition to the theoretical framework, this association between ACEs and negative health and well-being outcomes is mediated by the impact on one's stress system. Accumulating ACEs may dysregulate an individual's stress regulation and response system. Prolonged and excessive activation of the stress regulation and response system can lead to maladaptive responses to subsequent stressors and become ingrained in the child's neurobiological development [16,17]. When ingrained, it can elevate the risk of physical and mental health problems, and impair cognitive and social functioning, all of which may negatively affect well-being [17–19]. While considerable research exists on ACEs, stress impact, and overall well-being, the interplay among these factors is rarely examined [20].

Studies have indeed confirmed a higher incidence of ACEs in children with CP. ACEs have been studied mainly in DHH adults, but occasionally also in older DHH children and adolescents. Most studies researched the occurrence of a single ACE. For example, Bouldin et al [7] reported that DHH children are at increased risk of being bullied compared to their hearing peers. However, they may be less likely to engage in bullying behavior themselves. In another study, Kvam [21] identified a significantly elevated risk of sexual abuse among DHH children relative to hearing children and noted that attendance at special schools for the deaf constitutes an additional risk factor for abuse. In populations with DLD, ACEs appear to be understudied. The limited number of identified studies, all focusing on single ACEs, suggests that individuals with DLD may exhibit a comparable vulnerability to ACEs as observed in DHH populations. For example, Brownlie et al [8] found that women with

language impairments were more likely to report experiences of sexual abuse than those without language impairments. Similarly, two studies involving children and adolescents with DLD reported significantly higher levels of peer victimization than those reported by typically developing peers [22,23].

The occurrence of multiple ACEs has rarely been studied and only in DHH adults. Schenkel et al [24] compared the prevalence of various traumatic events, including some ACEs (emotional-, physical-, and sexual abuse, emotional-, and physical neglect), of 147 DHH college students to 317 hearing peers. The results revealed that DHH students reported significantly higher rates of abuse and neglect, as well as increased lifetime trauma exposure and post-traumatic stress disorder (PTSD) symptoms, relative to their hearing counterparts. Hall et al [25] examined ACE prevalence in a sample of 520 DHH adults aged 18 years and older, of whom 355 identified as DHH and 115 as deafblind. This study assessed the association between specific factors related to childhood hearing loss and the risk of encountering multiple ACEs. Results indicated that several factors, including less severe hearing loss, cochlear implant use, and not attending schools with access to sign language, were significantly associated with an increased likelihood of reporting two or more, as well as four or more, ACEs. Notably, Hall et al [25] 's findings contrast with those of Schenkel et al [24]; while Hall et al [25] identified less severe hearing loss as a risk factor for ACEs, Schenkel et al. [24] found that a larger degree of hearing loss was associated with increased risk of child maltreatment. Egbert [26] conducted a large-scale epidemiological study evaluating ACE prevalence among DHH and hearing adults. In this large-scale study, an ACE score of four or more was classified as indicative of high risk of toxic stress. The findings demonstrated no significant differences in ACE prevalence between DHH and hearing adults in the youngest (18–24 years) and oldest (65 years and older) age groups. However, among individuals aged 24–65, DHH adults showed a significantly higher likelihood of reporting high-risk ACE scores compared to hearing adults of the same age. In conclusion, studies by Schenkel et al [24] Hall et al [25], and Egbert [26] have reported accumulations of ACEs; however, these studies do not specify which particular ACEs are involved in these accumulations.

Therefore, to fully understand the unique vulnerabilities faced by DHH children and adolescents and those with DLD, it is crucial to conduct more nuanced and comprehensive research into the specific types of ACEs and their accumulation. Furthermore, so far, little attention has been given to the impact of these accumulated ACEs on stress and well-being. This underscores the need for research that involves a thorough assessment of childhood adversities (including their accumulation) experienced by DHH individuals and those with DLD, as well as the impact of these experiences on their stress levels and well-being.

## Current study

The present study focuses on ACEs, specifically in DHH adolescents and adolescents with DLD. This is because, during the dynamic period of adolescence, the adolescent brain appears to be more vulnerable to stress exposure [27]. During adolescence, individuals face increasing academic demands along with greater parental expectations for independence. At the same time, social demands intensify as peer interactions grow more complex. These rising expectations across academic, family, and social settings may cause increasing feelings of stress because challenges in language comprehension, communication modalities, and auditory processing may limit participation in class, interactions with parents and peers, and the ability to advocate for oneself [28–31]. Only adolescents with CP in special education will be assessed. In the Netherlands, students with CP who need help to develop their learning potential in mainstream education can receive support from a peripatetic teacher at their school. However, students with CP with educational needs that cannot be met in mainstream education qualify for special education.

The communication problems faced by DHH adolescents and adolescents with DLD appear to increase the risk of encountering similar childhood adversities; however, the origins of their communication problems may make them susceptible to different ACEs, at least in theory. To the best of our knowledge, no studies have directly compared the occurrence of ACEs between DHH children and adolescents and those with DLD. While on the one hand it could be hypothesized that similar patterns of ACEs might emerge across both groups because of early-life communication difficulties, one could on

the other hand argue that differences in the timing of early interventions, typically earlier in DHH populations [32,33], and in visible markers such as the use of sign language or hearing devices [34], may lead to distinct patterns of ACE exposure. Therefore, this study adopts a broad perspective, avoiding specific assumptions, to assess the prevalence and accumulation of ACEs while comparing these groups.

Finally, this research focuses on the relationship between the accumulation of ACEs, stress impact, and the well-being of adolescents. This is examined through a theoretical framework that posits stress as a mediator between adversity and well-being. Can the accumulation of ACEs be directly associated with reduced well-being in adolescents, or is it the stress evoked by ACEs that potentially affects adolescents' well-being? Individual differences in processing ACEs suggest that not all individuals exposed to multiple ACEs will exhibit elevated stress levels. As stress seems to play a critical role in determining an individual's physical and mental health [18], this study assumes that the impact of stress negatively affects the adolescents' well-being. In individuals with multiple accumulated ACEs where high stress is absent, the effect on well-being is assumed to be minimal or negligible. Hence, the assumption to be tested is that it is not the accumulation of ACEs, but rather the amount of stress evoked by the accumulated ACEs, that explains adolescents' well-being. If a mediating effect of the impact of stress is identified, it will be verified whether communication problems moderate this relationship.

In summary, this study seeks to answer the following questions: What is the prevalence and accumulation of ACEs (child abuse, household dysfunction, and additional social, economic and environmental ACEs) in adolescents with CP in special secondary education; Are there differences in ACE prevalence and accumulation between DHH adolescents and adolescents with DLD; Does the accumulation of ACEs predict the stress impact, and does the stress impact mediate the relationship between accumulated ACEs and the well-being of adolescents; and if this mediation is present, do the communication problems of the adolescents moderate this relationship?

## Method

### Ethics statement

This study received approval from the Ethics Committee of the Faculty of Social Sciences of Radboud University. Written, informed, and active consent was obtained from all participants to participate in this study. If a participant was under 16 years old, written, informed, and active consent was also obtained from their parents or guardians.

### Study design

For this cross-sectional study, a correlational research design was used to compare DHH adolescents with adolescents with DLD. Due to the absence of normed instruments for screening the incidence of ACEs, a reference group (RG) of adolescents without CP in mainstream secondary education was recruited for comparative purposes. The same correlational research design was applied to compare adolescents with CP to the reference group without CP. The study design included questionnaires addressing ACEs, stress impact, and well-being. An a priori power analysis was conducted using G*Power 3.1 [35], which indicated that per group, 28 participants were needed to detect a large effect size (Cohen's $d = .8$), with 90% power using a one-sided independent samples t-test with alpha at .05.

### Participants

The recruited participants with CP included DHH adolescents ($n = 32$, M age = 14.5 years, SD = 1.87, range: 12–17) and adolescents with DLD ($n = 95$, M age = 14.5 years, SD = 1.56, range: 12–17) who were attending special schools for secondary education. The reference group consisted of adolescents without CP from mainstream secondary schools ($n = 86$, M age = 14.2 years, SD = 1.66, range: 12–17) (see demographic details in Table 1).

**Table 1. Demographic Characteristics of Participants.**

| Participants | CP | RG | | | DHH | DLD | | |
|---|---|---|---|---|---|---|---|---|
| Demographics | % | % | *z* | Two-sided *p* | % | % | *z* | Two-sided *p* |
| **Gender** | | | | | | | | |
| Male | 62.99 | 62.79 | .030 | .976 | 65.63 | 62.11 | .357 | .721 |
| Female | 28.35 | 33.72 | -.836 | .403 | 25.00 | 29.47 | -.486 | .627 |
| Neutral/no ans. | 8.66 | 3.49 | 1.495 | .135 | 9.38 | 8.42 | .166 | .868 |
| **Education** | | | | | | | | |
| Vocational | 33.86 | 23.26 | -1.664 | .096 | 31.25 | 34.74 | .360 | .718 |
| Vocational (basic-advanced) | 52.76 | 10.47 | 6.321 | <.001** | 40.63 | 56.84 | -1.589 | .112 |
| General vocational | 9.45 | 25.58 | -3.154 | .002* | 18.75 | 6.32 | 2.080 | .038* |
| Senior general | 3.94 | 40.70 | 6.740 | <.001** | 9.38 | 2.11 | -1.829 | .067 |

Note: *N* = 213. Adolescents with CP, *CP n* = 127. Reference group, RG *n* = 86. DHH *n* = 32, DLD *n* = 95. *p* < .05. **p* < .001.

## Procedure

All 16 special secondary schools for students with CP in the Netherlands were approached to have their students participate in this study. Ten of these schools agreed to participate. In addition, seven mainstream secondary schools took part in this study. Participant recruitment occurred in classrooms from September 2, 2021, to September 28, 2022, through online video or in-person meetings, during which the first author explained the research project. Adolescents were offered 10 euros for their participation. The class mentor informed all parents via email or a paper letter with a link to an infographic and a video explaining the research project in Sign Supported Dutch and Sign Language of the Netherlands. The only exclusion criterion was a nonverbal IQ score of less than 80. The first author, a licensed and registered behavioral specialist, administered the data collection. Data collection took place between 2021 and 2023 and involved four student questionnaires. All questionnaires were read aloud in spoken Dutch and administered individually by the first author in a private room at each participating student's school. If DHH students received education supported with Sign Language of the Netherlands or Sign Supported Dutch, or when parents or students preferred, the test administration was supported by a hired, qualified, independent sign language interpreter. Before administering the questionnaires, students were informed of confidentiality rules. All information provided by the students was kept confidential, except in cases where there were signs of physical, emotional, or mental harm or distress. In these cases, professional standards required the first author to subsequently involve the school counselor and transfer responsibility for the student, following the protocol for domestic violence or child abuse. Within a month after participating, students and parents of students under 16 years received a report containing the results of the participant questionnaires, advice based on those results, contact information for professional healthcare services, and the first author's contact information for questions.

## Instrumentse

**General demographics.** General demographic information, including age, gender, hearing status, and educational attainment, was collected from the participants through a written questionnaire.

**Adverse childhood experiences.** ACEs were measured using a Dutch translation and adaptation of the ACE questionnaire developed by Felitti [36]. This quantitative instrument screens ten ACEs in two categories: Child Abuse and Household Dysfunction. The category Child Abuse consists of questions about emotional and physical abuse and neglect by a parent/caretaker and sexual abuse in general. The category Household Dysfunction refers to inter- and intra-parental problems. Because of growing evidence of the impact of socio-economic and community-level adversities on child development, additions to this ACE screener were recommended [37,38] and thus in this study six ACEs were

added to the questionnaire that are drawn from studies by Cronholm et al [37], Finkelhor et al [39], and Thurston et al [40]: bullying, discrimination, sickness or death of a loved one, experiencing an accident or disaster, and growing up in poverty. These six ACEs are unrelated and, hence, do not form a category. To evaluate the cumulative exposure to ACEs, this study calculates the total ACE score for adolescents by summing the items from the categories Child Abuse, Household Dysfunction, and the six additional ACEs. This created an ACE screener with 16 items in total, with answer options of 'yes' or 'no' (e.g., 'Were your parents ever separated or divorced?'), and scores ranging from 0 to 16.

**Stress impact.** The stress impact resulting from ACE(s) was measured with the Dutch translation of the Children's Revised Impact of Event Scale (CRIES-13, Child version) [41]. The CRIES-13 screens for the impact of a traumatic event. The CRIES-13 measures three symptoms of Post Traumatic Stress Disorder (PTSD): intrusions, avoidance, and arousal. Even though the questions of the CRIES-13 concerning a traumatic event are formulated singularly, the CRIES-13 can be used for screening PTSD based on multiple adverse events [42]. The CRIES-13 was administered after the ACE questionnaire was completed. As the questions refer to stressful events, the CRIES-13 was not administered to adolescents who reported zero ACEs. Adolescents who reported ACEs were instructed to answer the CRIES-13 questions about the ACEs, to which they responded 'yes'. The answer options required adolescents to rate 13 questions (e.g., 'Do you think about it, even when you don't want to?') on a 4-point Likert scale with scores: not at all (score 0), rarely (score 1), sometimes (score 3), often (score 5). The total score is a continuous score ranging from 0 to 65. Higher scores correspond with more PTSD symptoms. Additionally, a cutoff score of equal to or greater than 30 can serve as an indication of potentially having PTSD [43]. The internal consistency of the total scale is good (Cronbach's α 0.84) and acceptable for the subscales, intrusion, avoidance, and arousal (Cronbach's α 0.77, 0.72, 0.71) [43].

**Well-being.** Well-being was measured using the Dutch version of the Warwick-Edinburgh Mental Well-Being Scale (WEMWBS) [44]. The WENWBS comprises a single well-being scale consisting of 14 positively formulated statements about positive affect, satisfying interpersonal relationships, and positive functioning over the past two weeks. Adolescents rate these statements on a 5-point Likert scale, ranging from "none of the time" (score 1) to "all the time" (score 5), e.g., "I've been feeling useful." The total score is continuous, ranging from a minimum of 14 to a maximum of 70; a higher WEMWBS score indicates a higher level of mental well-being [44]. The WEMWBS exhibits high internal consistency (Cronbach's α = 0.91) and good test-retest reliability, with a mean of 0.86 (95% $CI$ = 0.82, 0.89) [45].

## Statistical analysis

Demographic differences were tested with proportion tests ($z$-tests). The main study outcome of the first two research questions was self-reported ACE prevalence and accumulation. All statistical analyses were performed using SPSS version 29 [46]. The descriptive statistics included percentages of categorical variables and means and standard deviations of continuous variables. Crosstabs, Chi-square tests for proportion comparison, and $t$-tests were applied to compare group prevalences and accumulations of ACEs. Outliers were kept, preserving the dataset's integrity and ensuring it accurately represents the data. Adolescents with CP were believed to be more vulnerable to ACEs related to their communication problems than the reference group; therefore, they were expected to have a higher prevalence and accumulation of ACEs, and the Chi-square analyses were conducted one-sided. The Chi-square analyses comparing DHH adolescents with adolescents with DLD were conducted two-sided. A Benjamini-Hochberg correction [47,48] was applied to all Chi-square analyses of the prevalences and accumulations of ACEs to address the multiple comparisons problem.

Before addressing the main study outcome of the third research question, $t$-tests were conducted to compare the stress impact and well-being between adolescents with CP, the reference group, DHH adolescents, and adolescents with DLD. ANCOVAs were performed, controlling for education (grouped into theoretical and practical attainment), comparing the reference group with adolescents with CP and DHH adolescents with adolescents with DLD, for the main group comparisons of ACE prevalence, stress impact, and well-being. A two-sided proportions test was used to compare the clinical scores on the stress impact screener of adolescents with CP and the reference group. To analyze the main study outcome

of the third research question, a simple mediation model, Hayes' process macro model 4 [49], was employed, using the continuous scores of the variables. The direct relationship between ACE accumulation and perceived well-being was analyzed, and the indirect effect of stress impact on the relationship between ACE accumulation and perceived well-being was tested. Additionally, a model 7 of the Hayes' process macro for moderated mediation [49] was employed to test the moderating effect of communication problems on the relationship between ACE accumulation and stress impact. Only the results of participants who completed the entire set of questionnaires were included in both models 4 and 7. For both models, a confidence interval that does not contain zero indicates a significant effect [49].

## Results

### Participant group differences

Table 1 provides detailed participant information, including gender distribution and educational attainment for the groups: adolescents with CP compared to the reference group, and DHH adolescents compared to adolescents with DLD. All groups were comparable in terms of these background characteristics, with a few exceptions. Significant differences were observed in the educational attainment of adolescents with CP compared to the reference group. A larger proportion of the reference group participated in a more theoretical educational attainment than adolescents with CP, while a larger proportion of adolescents with CP engaged in a more practical educational attainment than the reference group. Within the group of adolescents with CP, analysis showed that a significantly larger proportion of DHH adolescents participated in a more theoretical educational attainment compared to adolescents with DLD. After controlling for education as a possible covariate, no statistically significant differences were observed when comparing the reference group with adolescents with CP and DHH adolescents with adolescents with DLD in ACEs, stress impact, or well-being (see S1–S6 Tables).

### Adverse childhood experiences

An independent samples t-test was conducted to compare the prevalence of the ACE categories, Child Abuse and Household Dysfunction, as well as the total ACE prevalence of the 16 screened ACEs between adolescents with CP to the reference group. The results revealed that adolescents with CP reported significantly more Child Abuse than the reference group ($t_{(211)}$ = 3.76, one-sided $p < .001$). No differences were found between adolescents with CP and the reference group in experiencing Household Dysfunction ($t_{(211)}$ = 1.19, one-sided $p > .1$). Regarding total ACE prevalence, adolescents with CP reported significantly more ACEs compared to the reference group ($t_{(211)}$ = 3.06, one-sided $p < .001$) (see Table 2).

When comparing DHH adolescents and adolescents with DLD, an independent samples t-test was conducted to compare the prevalence of the ACE categories, Child Abuse and Household Dysfunction, as well as the total ACE prevalence of the 16 screened ACEs. No significant differences were found in experiencing Child Abuse ($t_{(125)}$ = -1.28, two-sided $p > .1$), and Household Dysfunction ($t_{(125)}$ = -1.05, two-sided $p > .1$). Also, the total ACE prevalence revealed no differences between DHH adolescents, and adolescents with DLD ($t_{(125)}$ = -1.07, two-sided $p > .1$) (see Table 2).

As shown in Table 3, a Chi-square test was used to compare adolescents with CP to the reference group to test for possible proportion differences within the categories of ACEs. A significantly larger proportion of adolescents with CP reported Child Abuse. Additionally, for all types of Child Abuse, the proportion of adolescents with CP was significantly larger than that of the reference group: emotional abuse ($p < .05$), physical abuse ($p < .001$), sexual abuse ($p < .05$), emotional neglect ($p < .05$), and physical neglect ($p < .05$). Regarding Household Dysfunction, only one significant difference was found. The proportion of adolescents with CP who reported having or having had a parent with a mental illness was larger compared to the reference group ($p < .05$). No differences between these groups were observed in terms of having a parent who was arrested/imprisoned, experiencing domestic violence, having a parent with a (history of) substance abuse, or having parents who divorced. As hypothesized, compared to the reference group, a larger proportion of adolescents with CP reported the additional ACE of being bullied ($p < .001$), but not the also hypothesized ACE of feeling

**Table 2. Means, Standard Deviations, and T-Tests of ACE Categories and Total ACE Prevalence.**

| Participants | CP | | RG | | One-sided p | t | 95% CI |
|---|---|---|---|---|---|---|---|
| ACEs | M | SD | M | SD | | | |
| Child abuse | 1.441 | 1.457 | .744 | 1.108 | <.001** | 3.759 | [.3, 1.1] |
| Household dysfunction | 1.402 | 1.335 | 1.186 | 1.242 | .118 | 1.189 | [-.1,.6] |
| Total (16) ACEs | 4.504 | 3.261 | 3.198 | 2.717 | .001* | 3.063 | [.5, 2.1] |
| Participants | DHH | | DLD | | Two-sided p | t | 95% CI |
| ACEs | M | SD | M | SD | | | |
| Child abuse | 1.156 | 1.322 | 1.537 | 1.493 | .202 | -1.282 | [-1.0,.2] |
| Household dysfunction | 1.188 | 1.091 | 1.474 | 1.405 | .296 | -1.049 | [-.8,.3] |
| Total (16) ACEs | 3.969 | 2.901 | 4.684 | 3.368 | .285 | -1.074 | [-2.0,.6] |

Note: $N = 213$. Adolescents with CP, CP $n = 127$. Reference group, RG $n = 86$. DHH $n = 32$, DLD $n = 95$. *$p < .05$. **$p < .001$.

**Table 3. Chi-Square ACE Prevalence Comparison.**

| | Participants % | | One-sided p | Participants % | | Two-sided p |
|---|---|---|---|---|---|---|
| ACEs | CP | RG | CP - RG | DHH | DLD | DHH - DLD |
| Child abuse | 63.78 | 41.86 | <.001** | 56.25 | 66.32 | .500 |
| Emotional abuse | 47.24 | 31.40 | .013* | 37.50 | 50.53 | .214 |
| Physical abuse | 38.58 | 15.12 | <.001** | 34.38 | 40.00 | 1.000 |
| Sexual abuse | 23.62 | 12.79 | .040* | 21.88 | 24.21 | 1.000 |
| Emotional neglect | 22.83 | 11.63 | .029* | 15.63 | 25.26 | .336 |
| Physical neglect | 11.81 | 3.49 | .022* | 6.25 | 13.68 | .312 |
| Household dysfunction | 67.72 | 63.95 | 1.000 | 65.63 | 68.42 | 1.000 |
| Parent(s) mental illness | 30.71 | 13.95 | .002* | 34.38 | 29.47 | 1.000 |
| Parent(s) arrested/imprisoned | 13.39 | 13.95 | 1.000ᴵ | 9.38 | 14.74 | .882 |
| Domestic violence | 48.82 | 36.05 | .066 | 40.63 | 51.58 | .425 |
| Parent(s) substance abuse | 16.54 | 9.30 | .167 | 15.63 | 16.84 | 1.000 |
| Parents divorced | 30.71 | 45.35 | .054ᴵ | 18.75 | 34.74 | .090 |
| Added ACEs | | | | | | |
| Bullied | 46.46 | 22.09 | <.001** | 40.63 | 48.42 | 1.000 |
| Discriminated | 31.50 | 24.42 | .393 | 37.50 | 29.47 | .716 |
| Experienced poverty | 15.75 | 8.14 | .115 | 9.38 | 17.89 | .284 |
| Severe disease loved one | 47.24 | 51.16 | 1.000ᴵ | 46.88 | 47.37 | 1.000 |
| Death caretaker/sibling | 5.51 | 3.49 | .889 | 9.38 | 4.21 | .371 |
| Accident/disaster | 19.69 | 17.44 | 1.000 | 18.75 | 20.00 | 1.000 |

Note: $N = 213$. Adolescents with CP, CP $n = 127$. Reference group, RG $n = 86$. DHH $n = 32$. DLD $n = 95$. *$p < .05$. **$p < .001$.ᴵTwo-sided p. Benjamini-Hochberg correction applied.

discriminated against ($p > .05$). No other proportion differences between these groups were found regarding the additional ACEs: experiencing poverty, having a loved one with a severe disease, experiencing the death of a parent or sibling, or experiencing an accident or disaster.

When comparing DHH adolescents to adolescents with DLD regarding categories of ACEs, no significant differences were found between DHH adolescents and those with DLD concerning Child Abuse and Household Dysfunction.

**Table 4. Chi-square ACE Accumulation Comparison.**

| ACE accumulation | Participants % | | One-sided p | Participants % | | Two-sided p |
|---|---|---|---|---|---|---|
| | CP | RG | CP - RG | D/HH | DLD | D/HH - DLD |
| ≥ 2 ACEs | 81.10 | 66.28 | .007* | 75.00 | 83.16 | .308 |
| ≥ 3 ACEs | 69.29 | 50.00 | .002* | 65.63 | 70.53 | .603 |
| ≥ 4 ACEs | 59.84 | 36.05 | <.001** | 53.13 | 62.11 | .370 |
| ≥ 5 ACEs | 40.94 | 25.58 | . 010* | 34.38 | 43.16 | .382 |
| ≥ 6 ACEs | 29.92 | 19.77 | .048* | 31.25 | 29.47 | .849 |
| ≥ 7 ACEs | 27.56 | 13.95 | .009* | 25.00 | 28.42 | .708 |

Note: $N = 213$. Adolescents with CP, CP $n = 127$. Reference group, RG $n = 86$. DHH $n = 32$. DLD $n = 95$. *$p < .05$. **$p < .001$. Benjamini-Hochberg correction applied.

Table 4 indicates that the proportion of adolescents with CP reporting accumulations of ACEs is significantly larger for all levels of ACE accumulation compared to those in the reference group ($p < .05$).

No significant differences were found between the proportions of DHH adolescents and adolescents with DLD in experiencing various levels of accumulation of ACEs.

## Stress impact

Because the stress impact questionnaire was administered only to adolescents who reported one or more ACEs, 23 participants were excluded from the analyses: reference group: $n = 10$, DHH adolescents: $n = 4$, and adolescents with DLD: $n = 9$. The stress impact questionnaire was completed by 190 adolescents (DHH $n = 28$, DLD $n = 86$, reference group $n = 76$). An independent samples $t$-test comparing adolescents with CP ($M = 30.12$, $SD = 14.81$) to the reference group ($M = 19.86$, $SD = 13.03$) revealed that adolescents with CP reported significantly higher stress impact than the reference group ($t_{(188)} = 4.92$, one-sided $p < .001$). Additionally, compared to the reference group, a significantly larger proportion of adolescents with CP (48.8%, $p < .001$) reported scores indicative of the potential for having PTSD (cutoff score 30, [43]; see S7, S8 Tables).

The independent samples $t$-test comparing the reported stress impact between the group of DHH adolescents and the adolescents with DLD revealed no significant difference.

## Well-being

When comparing adolescents with CP to the reference group on self-reported well-being, adolescents with CP reported significantly lower well-being ($M = 51.13$, $SD = 9.37$) than the reference group ($M = 54.69$, $SD = 9.44$) (see S9 Table, $t_{(211)} = -2.71$, one-sided $p < .05$).

In contrast, the independent samples $t$-test comparing DHH adolescents with adolescents with DLD on well-being showed no significant differences.

A simple mediation analysis was conducted using Hayes' process macro model 4 to test the indirect effect of stress impact on the relationship between ACE accumulation and adolescent well-being. The ACE and well-being questionnaires were completed by all participants ($n = 213$), while the stress impact questionnaire was completed only by those who reported one or more ACEs ($n = 190$). The mediation analysis was conducted using participants who completed all questionnaires ($n = 190$). It showed that ACEs had a significant direct positive effect on stress impact ($p < .001$). A bootstrap confidence interval for the indirect effect ($ab = -.468$), based on 5,000 bootstrap samples, was entirely below zero (-.761 to -.217) (see S10 Table). This suggests, as shown in Fig 1, that ACEs indirectly influence adolescents' well-being through their effect on stress impact. Adolescents who reported more ACEs ($a = 2.572$) also reported higher levels of perceived

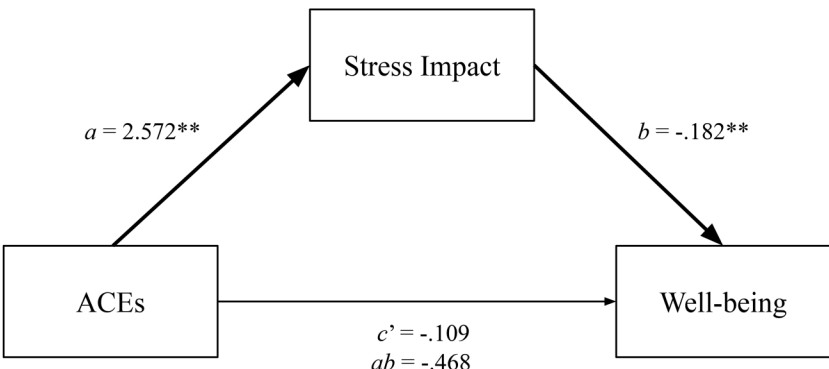

**Fig 1. Simple Mediation Model - Predicting Adolescents' Well-being.** Note: $N = 190$ (DHH $n = 28$, DLD $n = 86$, reference group $n = 76$). **$p < .001$. Statistics are standardized regression coefficients. Bold lines represent significant indirect paths. **$p < .001$.

stress (impact), resulting in lower reported well-being ($b = -.182$). When comparing the well-being of adolescents reporting zero ACEs, no differences were observed between adolescents with CP and the reference group (see S11 Table, $t_{(21)} = .102$, two-sided $p < .920$).

A moderated mediation analysis was conducted using Hayes' process macro model 7 (see S1 Fig) to analyze the moderating effect of having communication problems on the relationship between ACEs and stress impact in the simple mediation analysis. Having communication problems appears to have no moderating effect on the relationship between ACEs and stress impact (see S12 Table).

## Discussion

Results of this study indicate that adolescents with CP in special secondary education represent a vulnerable group regarding ACEs. The nature of their communication problems, whether being deaf or hard of hearing versus having DLD, does not appear to result in differences in vulnerability to ACEs. Adolescents with CP in special secondary education encounter significantly more ACEs than their peers without CP in mainstream secondary education and seem particularly susceptible to all types of Child Abuse. Furthermore, the prevalence of multiple ACEs (i.e., ≥ 2, ≥ 3, up to ≥ 7 ACEs) is significantly higher among adolescents with CP, with analyses demonstrating a strong association between ACE accumulation and perceived stress impact. The results of our mediation analysis appear to support the hypothesis that the prevalence of ACE negatively affects adolescents' lives by causing stress that adversely impacts their well-being.

Notably, nearly half of the adolescents with CP reported clinical scores on the CRIES-13, indicative of potential PTSD [43], suggesting this group might suffer substantial psychological stress. Mediation analyses supported the hypothesis that the negative effect of ACE accumulation on well-being operates indirectly through stress. Elevated stress levels were associated with lower self-reported well-being when considering the broader adolescent sample, including both the CP and reference groups. These findings suggest that it is not merely the accumulation of ACEs that matters, but rather the degree of stress they elicit that detracts from adolescents' well-being.

Differences in the origin of communication problems that may lead to different developmental challenges, such as the often prolonged period before DLD is diagnosed [33], or the visible markers associated with significant hearing loss (sign language or hearing devices) [34], do not appear to lead to variations in ACE exposure nor the stress impact and well-being in DHH adolescents and adolescents with DLD. Instead, it seems that communication challenges themselves may elevate the risk of ACEs, especially for all types of Child Abuse. This heightened vulnerability may result from difficulties in parent-child interactions related to impaired communication. While early detection of hearing loss followed by professional intervention is beneficial, it may not fully resolve difficulties in parent–child interactions. Having communication problems during childhood is

associated with an increased occurrence of challenging behavior in children and adolescents with CP [50,51], which may, in turn, exacerbate parental frustration [33,52,53]. Especially when children struggle to respond appropriately to verbal guidance, it may exacerbate parental stress and increase the risk of maltreatment. Caregivers may also interpret miscommunication as defiance, which may lead to inappropriate or punitive disciplinary strategies [25,54,55]. Moreover, parenting a child who develops differently than peers may lead to difficulties with acceptance and coping [56]. Considering the possibility that increased parental frustration within the parent-child dynamics may contribute to maltreatment, further research into this association is essential to better understand the relationship between having a child with CP, attending special education, and Child Abuse.

This study focused on adolescents with CP enrolled in special education. As a group, these adolescents differed from the reference group in mainstream education not only with respect to communication problems, but also with respect to educational attainment (possibly a proxy for learning ability) and having a parent with mental illness. It cannot be ruled out that these factors also contribute to vulnerability to ACEs. Moreover, these factors imply that the results of this study cannot be generalized to all students with communication problems, including those who attend regular schools.

Among the six additional ACEs evaluated, higher prevalences were expected among adolescents with CP for two specific adversities: being bullied and feeling discriminated against. Adolescents with CP reported significantly more experiences of being bullied than the reference group. This outcome aligns with previous research conducted among DHH children and adolescents [7,57], as well as among adolescents with DLD [22,23]. The adolescents with CP in this study attended special education. Although one might assume that placement in a special education setting among peers with similar needs would offer protection against bullying, studies in children and adolescents with CP indicate that special needs settings may offer less protective effect than envisaged [23,57]. Also, experiences of being bullied are not confined to school environments. It is possible that the surveyed adolescents with CP encountered challenges with social adjustment in contexts outside school, potentially increasing their vulnerability to being bullied. No differences were found between adolescents with CP and the reference group in reporting feelings of discrimination. On the one hand, it may be that, unlike being bullied, the special education setting may provide a protective buffer against perceived discrimination. The assessed adolescents with CP attended special secondary education alongside kindred peers, which may have mitigated feelings of being singled out or treated unfairly. Alternatively, the phrasing "Did you ever experience being treated differently/unpleasantly because of your appearance, skin color, religion, or culture?" may not have captured disability-related discrimination, thereby reducing the likelihood of differences between groups in response to this item.

Consistent with the present study, a systematic scoping review conducted by Lam et al [20] sought to expand the framework of adverse childhood experiences (ACEs) and their psychosocial outcomes by identifying potential links and mechanisms that might attenuate this association. Their review highlighted several studies that incorporated moderators, such as a child's race or sex assigned at birth, as well as mediators, including self-esteem and sleep problems. The broad range of identified moderating and mediating factors within the studies underscores the complexity of establishing a comprehensive framework. Although this body of work demonstrates considerable variability in the mediators explored, the findings of the current study emphasize that the stress elicited by ACEs may constitute a crucial mediating factor in the pathway linking ACEs to psychosocial outcomes.

As for the clinical relevance of this study, the detected prevalence and accumulation of the 16 ACEs reveal a rather alarming picture of adolescents with CP in special secondary education. These adolescents appear to be particularly susceptible to experiencing multiple types of Child Abuse and, notably, tend to accumulate more ACEs compared to peers without CP. Most concerning is that half of the participants (48.8%) reported ACE scores indicative of a potential risk for PTSD. In line with theoretical frameworks emphasizing the harmful effects of stress on health and well-being, this study indicates that stress is a more accurate predictor of adolescent well-being than the mere accumulation of ACEs. This highlights the importance of systematically screening adolescents with CP in special education for stress and well-being, with referrals to specialized healthcare, when necessary, as elevated stress levels can increase an individual's vulnerability to future stressors throughout their life [16,17]. Furthermore, in adults, a considerable amount of time has elapsed between

the occurrence of ACEs and the moment they are assessed. If ACEs have had a negative impact, interventions may come too late to prevent long-term consequences. Earlier identification and intervention are therefore essential.

## Limitations and recommendations

This study focused on adolescents enrolled in special secondary education. This limits the generalizability of the outcomes to the broader population of adolescents with CP. Given the severity of the outcomes observed, further research is warranted to determine whether similar risks exist among students with CP in mainstream education or if this subgroup constitutes an exception.

The seriousness of the results calls for actions aimed at preventing ACEs in children and adolescents with CP in special education. Systematic psychological health screening during adolescence and childhood appears to be a crucial first step. In addition, further research is needed to develop and evaluate interventions aimed at mitigating the stress associated with ACEs to reduce their detrimental effects on well-being. Exploring the role of resilience in children and young people in this context may also serve as an essential starting point for future investigations.

Although increased vulnerability to various ACEs in youth with CP seems evident, little is still known about the factors that contribute to this heightened vulnerability. Further research is needed on the elements of the parent-child dynamic that may contribute to the increased risk of physical and emotional abuse or neglect. Additionally, the factors that make youth with CP vulnerable to sexual abuse and being bullied could be examined.

## Supporting information

**S1 Table. ACE Total Tests of Between-Subjects Effects Reference Group - Target Group.**
(PDF)

**S2 Table. Stress Impact Tests of Between-Subjects Effects Reference Group - Target Group.**
(PDF)

**S3 Table. Well-being Tests of Between-Subjects Effects Reference Group - Target Group.**
(PDF)

**S4 Table. ACE Total Tests of Between-Subjects Effects DHH Adolescents - Adolescents with DLD.**
(PDF)

**S5 Table. Stress Impact Tests of Between-Subjects Effects DHH Adolescents - Adolescents with DLD.**
(PDF)

**S6 Table. Well-being Tests of Between-Subjects Effects DHH Adolescents - Adolescents with DLD.**
(PDF)

**S7 Table. T-Test Comparing Stress Impact.**
(PDF)

**S8 Table. Independent Samples Proportion Test Clinical Scores Stress Impact (PTSD Screener CRIES-13).**
(PDF)

**S9 Table. T-Test Comparing Well-being.**
(PDF)

**S10 Table. Mediation Effect of Stress Impact on Well-Being.**
(PDF)

**S11 Table. T-Test Comparing Groups Reporting Zero ACEs on Well-being.**
(PDF)

**S12 Table. Moderated Mediation - Moderation Effect of Communication Problems on Mediation ACEs, Stress Impact, and Well-being.**
(PDF)

**S13 Table. Comparing Total ACE Prevalence DHH Adolescents - Adolescents with DLD.**
(PDF)

**S14 Table. Comparing Total ACE Prevalence Adolescents with CP - Reference Group.**
(PDF)

**S15 Table. Independent Samples Proportion Tests Sexual Abuse.**
(PDF)

**S16 Table. Robust Test of Equality of Means Child Abuse, Household Dysfunction, ACE Total, Stress Impact, Well-being, Reference Group - Target Group.**
(PDF)

**S17 Table. Robust Test of Equality of Means Child Abuse, Household Dysfunction, ACE Total, Stress Impact, Well-being, DHH Adolescents - Adolescents with DLD.**
(PDF)

**S18 Table. Independent Samples Effect Sizes Child Abuse, Household Dysfunction, ACE Total, Stress Impact, Well-being, Reference Group - Target Group.**
(PDF)

**S19 Table. Independent Samples Effect Sizes Child Abuse, Household Dysfunction, ACE Total, Stress Impact, Well-being, DHH Adolescents - Adolescents with DLD.**
(PDF)

**S1 Fig. Moderated Mediation Model.**
(TIF)

## Acknowledgments

The authors wish to thank all the youth who participated in this study.

## Author contributions

**Conceptualization:** Len Martijn, D. Hermans, H.E.T. Knoors, C.T.W.M. Vissers.

**Data curation:** Len Martijn.

**Formal analysis:** Len Martijn.

**Funding acquisition:** Len Martijn.

**Investigation:** Len Martijn.

**Methodology:** Len Martijn, D. Hermans, H.E.T. Knoors, C.T.W.M. Vissers.

**Project administration:** Len Martijn.

**Supervision:** D. Hermans, H.E.T. Knoors, C.T.W.M. Vissers.

**Writing – original draft:** Len Martijn.

**Writing – review & editing:** D. Hermans, H.E.T. Knoors, C.T.W.M. Vissers.

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
