## [Decision Letter · Decision Letter 0]

22 Aug 2025

PMEN-D-25-00299

Adverse childhood experiences, stress impact, and well-being in deaf and hard of hearing adolescents and adolescents with developmental language disorders in special secondary education

PLOS Mental Health

Dear Dr. Martijn,

Thank you for submitting your manuscript to PLOS Mental Health. After careful consideration, we feel that it has merit but does not fully meet PLOS Mental Health’s publication criteria as it currently stands. Therefore, we invite you to submit a revised version of the manuscript that addresses the points raised during the review process.

We look forward to receiving your revised manuscript.

Kind regards,

Lambert Zixin Li, Ph.D.

Academic Editor

PLOS Mental Health

Journal Requirements:

1. Please provide additional details regarding participant consent. In the ethics statement in the Methods and online submission information, please ensure that you have specified (1) whether consent was informed and (2) what type you obtained (for instance, written or verbal, and if verbal, how it was documented and witnessed). If your study included minors, state whether you obtained consent from parents or guardians. If the need for consent was waived by the ethics committee, please include this information.

2. Please send a completed 'Competing Interests' statement, including any COIs declared by your co-authors. If you have no competing interests to declare, please state "The authors have declared that no competing interests exist". Otherwise please declare all competing interests beginning with the statement "I have read the journal's policy and the authors of this manuscript have the following competing interests:"

3. Please amend your detailed Financial Disclosure statement. This is published with the article. It must therefore be completed in full sentences and contain the exact wording you wish to be published.

1. Please clarify all sources of funding (financial or material support) for your study. List the grants (with grant number) or organizations (with url) that supported your study, including funding received from your institution. 

2. State the initials, alongside each funding source, of each author to receive each grant.

3. State what role the funders took in the study. If the funders had no role in your study, please state: “The funders had no role in study design, data collection and analysis, decision to publish, or preparation of the manuscript.”

4. If any authors received a salary from any of your funders, please state which authors and which funders.

4. Please note that your Data Availability Statement is currently missing the repository name. If your manuscript is accepted for publication, you will be asked to provide these details on a very short timeline. We therefore suggest that you provide this information now, though we will not hold up the peer review process if you are unable.

5. We have noticed that you have a list of Supporting Information legends in your manuscript. However, there are no corresponding files uploaded to the submission. Please upload them as separate files with the item type 'Supporting Information'.

Additional Editor Comments:

Thank you for submitting your manuscript to PLOS Mental Health. You have chosen an important and timely topic. We kindly ask that you address the reviewers’ suggestions, which I will not repeat here, before we can reconsider your manuscript for publication.

Reviewers' comments:

Reviewer's Responses to Questions

**Comments to the Author**

1. Does this manuscript meet PLOS Mental Health’s publication criteria?

Reviewer #1: Partly

Reviewer #2: Yes

2. Has the statistical analysis been performed appropriately and rigorously?

Reviewer #1: Yes

Reviewer #2: No

3. Have the authors made all data underlying the findings in their manuscript fully available (please refer to the Data Availability Statement at the start of the manuscript PDF file)?

Reviewer #1: Yes

Reviewer #2: Yes

4. Is the manuscript presented in an intelligible fashion and written in standard English?

Reviewer #1: Yes

Reviewer #2: Yes

Reviewer #1: This paper highlights a significant findings of a vulnerable population. Congratulations for the authors for prsenting the research findings. However, I would like to share my observations which needs to be addressed.

1. The paper lacks a clear and scientifically sound methodology which can substantiate the quality of the study.

2. The paper lacks a theoretical framework for the study.

3. There is no proper description about the researchers qualification, sampling and data collection process.

4. Discussion section lacks support/ refute of other research adequately.

5. This paper needs to be included more recent citations.

All the best.

Reviewer #2: Great topic and timely paper—linking ACEs with adolescent CP in DHH/DLD is important and under-studied. For framing, a brief addition to the Introduction would suffice: 1. add 2–3 sentences summarizing developmental considerations for DHH and DLD in adolescence (e.g., rising social/academic demands, later identification in DLD, visibility and accommodation issues in DHH), and state whether you conceptualize their impact on CP as developmentally stable across ages 12–17 or likely to intensify with age. Given the cross-sectional design, I’m not asking for new analyses—just make the assumption explicit and note the design limitation.

2. ACE Composition — Justify aggregating child abuse, household dysfunction, and added community/socioeconomic items into one ACE index. Report category-level results and run sensitivity analyses (abuse-only, household-dysfunction-only, ACE total excluding household dysfunction) to test mediation robustness.

3. Covariate Control — Since the three groups differ demographically, include key covariates in main comparisons and in mediation/moderation models (e.g., age, sex, educational track, living with family; consider school clustering).

4. Statistical Robustness — With unbalanced n and likely heteroscedasticity, use Welch’s t (or preferably linear models with HC3 robust SEs and planned contrasts). Report Hedges’ g with CIs, add robust/nonparametric checks (trimmed-mean t, permutation/bootstrap), and re-estimate mediation/moderation with robust SEs plus covariates.

**Do you want your identity to be public for this peer review?** For information about this choice, including consent withdrawal, please see our Privacy Policy

Reviewer #1: No

Reviewer #2: No

---

## [Editor Report · Decision Letter 1]

4 Nov 2025

Adverse childhood experiences, stress impact, and well-being in deaf and hard of hearing adolescents and adolescents with developmental language disorders in special secondary education

PMEN-D-25-00299R1

Dear MSc Martijn,

We are pleased to inform you that your manuscript 'Adverse childhood experiences, stress impact, and well-being in deaf and hard of hearing adolescents and adolescents with developmental language disorders in special secondary education' has been provisionally accepted for publication in PLOS Mental Health.

Best regards,

Lambert Zixin Li, Ph.D.

Academic Editor

PLOS Mental Health

Dear authors,

Thank you for revising your manuscript and responding carefully to the reviewers’ comments. I have reviewed your revised paper and response memo, and find the revisions satisfactory. The manuscript is now of publishable quality. Congratulations, and thank you for choosing our journal for your work.

Kind regards,

Lambert Zixin Li, PhD